# Standardized Diagnostic Assays for Omsk Hemorrhagic Fever Virus

**DOI:** 10.3390/pathogens14111093

**Published:** 2025-10-27

**Authors:** Jeong-Hyun Lee, Sunyoung Jung, Hwajung Yi, Yoon-Seok Chung

**Affiliations:** Division of High-Risk Pathogens, Department of Laboratory Diagnosis and Analysis, Korea Disease Control and Prevention Agency (KDCA), Cheongju 28159, Republic of Korea

**Keywords:** Omsk hemorrhagic fever virus, real-time reverse transcription polymerase chain reaction assay, diagnostic assay for Omsk hemorrhagic fever virus, internal control, human ribonuclease P, standard operating procedure, outbreak preparedness, biosurveillance

## Abstract

Omsk hemorrhagic fever is an acute zoonotic disease caused by Omsk hemorrhagic fever virus, a member of the genus Flavivirus (family Flaviviridae), with a reported case-fatality rate of approximately 3%. Historically confined to southwestern Siberia, ecological changes raise concerns about possible spread to non-endemic regions. Although no Omsk hemorrhagic fever cases have been reported in the Republic of Korea, the risk of accidental importation highlights the importance of establishing a reliable diagnostic protocol. We established and validated an institutionally developed diagnostic protocol employing real-time reverse transcription polymerase chain reaction targeting the *NS2A* and *C* genes of Omsk hemorrhagic fever virus. Primers and probes were designed from all available genomes to ensure broad strain coverage. Human ribonuclease P was used as an internal control to verify nucleic acid extraction and amplification. Using synthetic deoxyribonucleic acid fragments and in vitro-transcribed ribonucleic acid, assay performance was optimized, and analytical sensitivity was determined using probit analysis. The limits of detection were 74.50 copies/µL (threshold cycle 32.99) for *NS2A* and 70.41 copies/µL (threshold cycle 35.38) for *C*. Specificity testing using representative flaviviruses (West Nile virus, Yellow fever virus, Zika virus, St. Louis encephalitis virus, and Tick-borne encephalitis virus) and an alphavirus (Venezuelan equine encephalitis virus) demonstrated no cross-reactivity. The assay demonstrated high sensitivity, specificity, and reproducibility, supporting its potential application in national and international Omsk hemorrhagic fever virus surveillance systems.

## 1. Introduction

Omsk hemorrhagic fever (OHF) is a zoonotic disease caused by Omsk hemorrhagic fever virus (OHFV), a member of the genus Flavivirus within the family Flaviviridae [1]. The disease was first described in the Omsk region of southwestern Siberia in the 1940s [2,3]. Transmission occurs primarily through rodents and ticks, with humans acquiring the infection via tick bites, contact with contaminated water, or handling of infected rodents [1,3]. OHFV is antigenically and structurally related to tick-borne encephalitis virus (TBEV), making differential diagnosis based on clinical presentation challenging [4,5].

The clinical spectrum of OHF ranges from nonspecific febrile illness with headache, chills, and vomiting to hemorrhagic signs and severe neurological complications, with a reported case-fatality rate of 0.5–3% [1]. With the absence of vaccines or specific antiviral therapies, OHFV is classified as a biosafety level-4 (BSL-4) pathogen [6,7], underscoring its significance as a high-consequence infectious disease agent. Although OHF remains geographically restricted to southwestern Siberia, recent ecological changes and an increase in human mobility have raised concerns about possible spread to non-endemic regions [2,3]. To date, no cases have been reported in ROK; nevertheless, the risk of accidental importation highlights the urgent need for preparedness by establishing reliable diagnostic protocols [1,2].

Current diagnostic methods include serological assays and antigen detection; however, they are limited by cross-reactivity with other flaviviruses, particularly TBEV [5]. Virus isolation, while highly specific, requires BSL-4 containment and is impractical for routine laboratory use [6]. Molecular diagnostics, especially real-time reverse transcription polymerase chain reaction (real-time RT-PCR), have become the most sensitive and specific approach for OHFV detection [8,9]. One such system, a SYBR Green I-based RT-PCR assay for OHFV detection, has been developed and characterized [4], providing a methodological foundation for subsequent improvements in molecular diagnosis. Nevertheless, most existing assays have not been fully validated or standardized for diagnostic use due to the limited availability of clinical specimens [1,10].

Taken together, establishing a standardized and reproducible molecular diagnostic assay for OHFV is urgently needed. In this study, we designed new primers and probes targeting the *NS2A* and *C* genes of OHFV, incorporated human RNase P as an internal control (IC), and validated assay performance using synthetic DNA fragments and in vitro-transcribed RNA. Importantly, this assay has now been integrated into an institutionally developed diagnostic protocol in ROK, ensuring preparedness for potential OHFV incidence and contributing to global biosurveillance and outbreak response capacity.

## 2. Materials and Methods

### 2.1. Specimens and Reagents

#### 2.1.1. Specimens [5,7,8]

Whole blood and serum are the primary specimens, and cerebrospinal fluid (CSF) is the secondary specimen. Although blood specimens are generally recommended, CSF is considered when neurological symptoms are present or when clinically indicated. For blood sample collection, whole blood and serum samples (≥4 mL) are collected in ethylene diamine tetraacetic acid (EDTA)-treated tubes or serum separation tubes, while heparin-treated tubes are avoided due to potential inhibition of PCR amplification [9]. CSF samples (1.5–2 mL) are collected in sterile containers. Specimens are obtained from suspected individuals with OHF at the onset of symptoms. All samples are packaged according to the “Guidelines for the Safe Transport of Infectious Substances,” using triple packaging and are stored and transported under refrigerated conditions (2–8 °C) within 24 h of collection. Although this assay was originally designed for testing human clinical specimens, no human or animal OHFV infections have been reported in Korea to date. Therefore, analytical validation was performed using in vitro-transcribed RNA standards instead of clinical samples.

#### 2.1.2. Controls

Positive controls were prepared as synthetic RNA corresponding to the OHFV *NS2A* and *C* genes (GenBank No.: NC_005062). Based on the reference sequences, synthetic RNA was generated by ATOPLEX Co., Ltd. (Hanam-si, Gyeonggi-do, Republic of Korea), diluted in DNase/RNase-free water to a working concentration of 1 × 10^6^ copies/μL, and used for assay validation. Negative controls consisted of DNase/RNase-free water, which was included in each assay to monitor nonspecific amplification and potential contamination.

The human ribonuclease P (hRNase P) gene was used as the IC. Synthetic hRNase P RNA obtained from Integrated DNA Technologies, Inc. (Coralville, IA, USA) was prepared at a concentration of 1 × 10^3^ copies/μL and was co-amplified with OHFV synthetic RNA to ensure assay reliability.

#### 2.1.3. Commercial Reagents

For viral RNA extraction, the QIAamp Viral RNA Mini Kit (QIAGEN, Hilden, Germany) was used and stored at 15–25 °C. For nucleic acid amplification, both the Custom TaqMan™ Gene Expression Assay (Applied Biosystems, Waltham, MA, USA), consisting of a primer–probe set, and the TaqPath™ 1-Step RT-qPCR Master Mix (Applied Biosystems, Waltham, MA, USA), stored at −30 to −10 °C, were used. Additional reagents included UltraPure Distilled Water (Invitrogen, Waltham, MA, USA), free of DNase and RNase, stored at 15–25 °C; absolute ethanol, biotechnology grade (Merck, Darmstadt, Germany), stored at 15–25 °C; and Micro-Chem Plus disinfectant detergent (National Chemical Laboratories, Philadelphia, PA, USA), stored at 10–48 °C.

#### 2.1.4. Primer/Probe Sequences [5,6,7]

Primers and probes targeting the OHFV *NS2A* and Core Protein *C* genes, as well as the hRNase P gene (internal control), were synthesized based on reference nucleotide sequences and were custom-ordered for use in this study.

For the OHFV *NS2A* gene, the forward primer (OHFV1-F: 5′-ATCCCAGAATGGTGCTGC-3′), reverse primer (OHFV1-R: 5′-ACTGGCCGTATCTCCATG-3′), and probe (OHFV1-P: 5′-FAM-AGTCAGTCCCTGTTCTGAATGTCACCG-QSY-3′) were used. For the OHFV Core Protein *C* gene, the forward primer (OHFV2-F: 5′-CTCGACGAGTGTCGAAAGA-3′), reverse primer (OHFV2-R: 5′-CTCCATTATGCGCTTCAACA-3′), and probe (OHFV2-P: 5′-VIC-TGGTCCAAATGCCAAATGGA-QSY-3′) were applied. As the internal control, primers and probe targeting the hRNase P gene were used: forward primer (hRNase P-F: 5′-AGATTTGGACCTGCGAGCG-3′), reverse primer (hRNase P-R: 5′-GAGCGGCTGTCTCCACAAGT-3′), and probe (hRNase P-P: 5′-Cy5-TTCTGACCTGAAGGCTCTGCGCG-QSY2-3′).

### 2.2. Biosafety Considerations

Clinical specimen pre-treatment is performed in a BSL-4 laboratory within a Class II biosafety cabinet, and subsequent RNA extraction and RT-PCR reagent preparation are conducted in a BSL-2 laboratory. All procedures are performed in compliance with the Korea Disease Control and Prevention Agency (KDCA) Laboratory Biosafety Guidelines (2025), and operators wear appropriate personal protective equipment. Laboratory environments are disinfected after use, and emergency preparedness, including access to fire extinguishers, spill kits, and first-aid equipment, is maintained.

### 2.3. Assay Implementation

#### 2.3.1. Specimen Pretreatment [10,11,12]

Clinical specimens obtained from suspected individuals with OHF are processed in the BSL-4 laboratory of the KDCA in accordance with national biosafety regulations. Following inactivation, the specimens are transferred to the BSL-2 laboratory for downstream molecular analyses.

#### 2.3.2. RNA Extraction

RNA extraction is performed using the QIAamp Viral RNA Mini Kit (Qiagen, Hilden, Germany) according to the manufacturer’s protocol. The inactivated specimen mixture (1260 µL) is briefly centrifuged at 8000 rpm for 10 s, and 630 µL is loaded onto the QIAamp spin column twice, with centrifugation at 8000 rpm for 1 min each. The column is then washed with 500 µL of Buffer AW1, followed by centrifugation at 8000 rpm for 1 min, with an additional wash for blood specimens. It is subsequently washed with 500 µL of Buffer AW2 and centrifuged at 14,000 rpm for 3 min. After a final dry spin, RNA is eluted with 60 µL of Buffer AVE and stored at −20 °C or −80 °C.

#### 2.3.3. PCR Mixture Preparation and RNA Addition

Before PCR setup, the workspace and pipette surfaces were disinfected with 70% ethanol. The PCR mixture was prepared, and 15 µL of the mixture was dispensed into each real-time PCR reaction tube. Subsequently, 5 µL of extracted RNA or positive/negative control was added to each tube. The tubes were tightly sealed and briefly centrifuged to collect the reaction mixture at the bottom.

Each reaction had a final volume of 20 µL, consisting of 1 µL of Custom MultiPlex TaqMan Assay (20×), 5 µL of TaqPath™ 1-Step RT-qPCR Master Mix, CG, 9 µL of nuclease-free water, and 5 µL of RNA template or control. In practice, 15 µL of master mix (1 µL Assay + 5 µL Master Mix + 9 µL water) was dispensed into each tube, followed by 5 µL of extracted RNA or positive/negative control to reach a total reaction volume of 20 µL. 

#### 2.3.4. Real-Time RT-PCR Assay

The assays were performed using the QuantStudio™ Dx Real-Time PCR System (Applied Biosystems, Waltham, MA, USA), and data were analyzed with QuantStudio™ Dx Software v1.3. Fluorescence detection was carried out through the FAM, VIC, and Cy5 channels, with the baseline automatically set. The thresholds were fixed at 0.1 ΔRn for NS2A and C targets and 0.2 ΔRn for the hRNase P to determine Ct values.

RT-PCR amplification was performed in a total volume of 20 μL. The thermal cycling conditions were as follows: incubation at 25 °C for 2 min, reverse transcription at 50 °C for 15 min, and polymerase activation at 95 °C for 2 min, followed by 40 cycles of denaturation at 95 °C for 3 s and annealing/extension at 60 °C for 30 s, with fluorescence measured at 56 °C during amplification.

#### 2.3.5. Result Interpretation

Ct values for each target gene were interpreted according to predefined threshold rules. Fixed ΔRn thresholds were empirically determined by evaluating baseline windows across 10-fold serial dilutions of synthetic OHFV RNA and NTC runs; settings that minimized background and NTC noise while preserving linearity within the validated range were selected. For analytical runs, the baseline was automatically set, and thresholds were fixed at 0.1 ΔRn for NS2A and C, and 0.2 ΔRn for the hRNase P to determine Ct values.

## 3. Results

For performance evaluation, cutoff criteria were established through preliminary testing, and inconclusive ranges were determined using Probit analysis. Specificity was assessed through positive and negative agreement testing, while precision was evaluated to confirm repeatability and reproducibility. Cross-reactivity was addressed within the specificity assessment, and interference studies further ensured analytical reliability.

### 3.1. Establishment of Cutoff Thresholds

A specimen was judged positive when the Ct value of the *NS2A* gene was <34.05 and that of the *C* gene was <36.24 (Appendix A, Case 1). A specimen was judged negative when the Ct value of the *NS2A* gene was >35.04 and that of the *C* gene was >37.04, or when both targets were undetected (Appendix A, Case 3). Results were considered inconclusive under the following conditions, which require repeat testing: (i) Ct values of *NS2A* were 32.99–35.04 and those of the *C* gene were 35.38–37.04 (Appendix A, Cases 2, 4, 5–10); (ii) positive or negative controls were invalid, requiring the replacement of operator, reagents, or controls and repeat RNA extraction (Appendix A, Cases 11–13). If inconclusive results persisted even after repeat testing, specimens were recollected and testing was repeated. For the final judgment, when more than one specimen type was submitted for OHFV, the case was considered positive if any specimen type tested positive.

For specimens that were initially positive for only one target gene (NS2A or C), we applied a predefined re-test/confirmation workflow. Nucleic acids were re-extracted, and the specimen was re-tested in duplicate for both targets in the same run. A result was deemed confirmed positive only when the initially positive target was detected in both duplicate wells within the positive zone. With a valid internal control (IC), duplicate-negative results were classified as negative. Results that remained within the inconclusive zone or showed positivity in only one duplicate were reported as inconclusive, and recollection was recommended. IC failure or control failure (PC/NC) rendered the run invalid, prompting repeat testing from RNA re-extraction (replacing operator/reagents/controls as needed). When strong target signals suppressed IC amplification, specimens were diluted and re-tested (cutoff definitions are provided in Appendix A).

### 3.2. Preliminary Testing

The specimen list and results for the preliminary testing are summarized in Appendix A.

### 3.3. Linearity and Amplification Efficiency

The standard curves were generated through simple linear regression of Ct versus log_10_(input copies) using a 10-fold dilution series from 10^2^ to 10^5^ copies/μL. The slopes were −3.697 (NS2A) and −3.709 (C), with coefficients of determination (R^2^) of 0.9998 and 0.9999, respectively. Amplification efficiency was calculated as E (%) = (10⁽^−^^1^/ˢˡᵒᵖᵉ⁾ − 1) × 100, yielding 86.4% for NS2A and 86.0% for C. Data at 10^1^ copies/μL were excluded from the linearity assessment due to partial detection for NS2A and non-detection for C; therefore, the linear dynamic range was defined as 10^2^–10^5^ copies/μL for both targets. Representative amplification and standard-curve plots are provided in Appendix A.

### 3.4. Inconclusive Zone and Limit of Detection (LOD)

In the preliminary testing, consistent amplification was observed at concentrations of 1 × 10^5^ to 1 × 10^2^ copies/µL with 100% detection, while no amplification was detected at ≤1 × 10^1^ copies/µL. Inconclusive zones showed sporadic amplification, and no OHFV-specific signals were observed in negative controls. The IC was reliably amplified in all valid reactions.

### 3.5. Probit Analysis Results

For the Probit analysis, both *NS2A* and *C* genes were consistently detected at concentrations of 80–100 copies/µL with a detection rate of 100%. Detection rates gradually decreased to 70–80% at 60 copies/µL and 20–30% at 40 copies/µL, while no amplification was observed at ≤20 copies/µL. Probit analysis estimated the C_50_ values at 50.24 copies/µL (95% CI: 41.37–58.16) for *NS2A* and 46.44 copies/µL (95% confidence interval (CI): 36.88–54.21) for *C*. Positive controls were consistently amplified, negative controls showed no OHFV-specific amplification, and the IC (hRNase P) was reliably detected in all valid reactions. The detailed experimental results are presented in Appendix A.

### 3.6. Results for Cutoff Threshold Determination

In the cutoff threshold determination, mean Ct values of 34.05 for NS2A at 50.24 copies/µL (25/40 replicates) and 36.24 for C at 46.44 copies/µL (23/40 replicates) were established, which served as the cutoff thresholds for each target.

### 3.7. Establishment of Inconclusive Zone

In the inconclusive zone determination, the upper and lower Ct boundaries for the *NS2A* gene were confirmed as 32.99 (74.50 copies/µL) and 35.04 (33.88 copies/µL), respectively, while those for the *C* gene were confirmed as 35.38 (70.41 copies/µL) and 37.04 (30.63 copies/µL), respectively (Appendix A).

The detection limits were determined using serial dilutions of synthetic OHFV RNA, from which threshold values were empirically established based on fluorescence baseline and exponential curve analysis. The selected parameters provided the most consistent Ct values across replicates, minimizing background noise. It should be noted that these detection limits were derived from synthetic RNA targets, and actual limits may vary in biological samples due to potential matrix effects (e.g., inhibitors in serum or cerebrospinal fluid). This procedure was performed in accordance with WHO and CDC guidelines for qPCR assay standardization.

### 3.8. Positive and Negative Percent Agreement

Diagnostic accuracy was evaluated by comparing assay results with the reference standard. Among 100 contrived specimens (50 positive and 50 negative), all OHFV-positive samples were correctly detected (50/50), and all negative samples were non-reactive (50/50), yielding a positive percent agreement (PPA) of 100% (95% CI, 92.9–100.0%) and a negative percent agreement (NPA) of 100% (95% CI, 92.9–100.0%). The gold standard was defined as the true positive or negative status of each specimen. As no domestic cases of OHFV have been reported, clinical specimens were unavailable, and positive contrived specimens were prepared based on the LOD results (Appendix A). Among 100 evaluated specimens, all OHFV-positive samples consistently amplified the NS2A and C targets, confirming assay sensitivity, whereas negative samples containing viral and bacterial nucleic acids showed no OHFV-specific amplification, demonstrating specificity. The IC (hRNase P) was reliably amplified in all reactions, verifying the validity of nucleic acid extraction and PCR performance. Overall, the assay achieved 100% PPA and 100% NPA compared with reference results, indicating high diagnostic accuracy. Inclusion of positive and negative controls further supported assay validity. Detailed Ct values and interpretation outcomes are summarized in Appendix A.

### 3.9. Precision (Repeatability and Reproducibility)

For precision evaluation, four specimens were prepared. Three specimens consisted of synthetic RNAs targeting *NS2A* and *C*, formulated at concentrations corresponding to the performance assessment levels of 1.5× LOD, 3× LOD, and 10× LOD. Quantification through real-time PCR yielded 111.75 copies/µL (*Ns2a*) and 106.51 copies/µL (*C*) at 1.5× LOD; 223.5 and 211.23 copies/µL at 3× LOD; and 745 and 704.1 copies/µL at 10× LOD, respectively. The negative specimen consisted of normal human serum (Sigma). As an IC, the hRNase P (1 × 10^3^ copies/µL) was incorporated during specimen preparation. This panel encompassed concentrations from near the LOD to higher input levels, confirming the expected detection of both targets in positive specimens and no detection in the negative matrix, thereby supporting the assay’s precision assessment. Operators A and B performed repeated gene detection tests on three OHFV-positive specimens and one negative specimen over a period of 5 days. All test results were concordant, and repeatability and reproducibility were confirmed to be 100%. The detailed repeatability and reproducibility results for Operator A and Operator B are provided in Appendix A.

### 3.10. Cross-Reactivity and Interference

Cross-reactivity testing was not performed separately; instead, it was replaced with a specificity analysis. Specificity was evaluated using PPA and NPA analysis and further verified through an in silico analysis of most viral genomes, followed by specificity testing using a variety of RNA and DNA materials. Cross-reactivity was evaluated using synthetic RNAs from closely related flaviviruses (WNV, YFV, Zika [Asian lineage], SLEV, TBEV) and a representative alphavirus (VEEV); no OHFV-specific amplification was observed. However, because testing used synthetic RNAs rather than clinical matrices and did not include DENV, the possibility of matrix-dependent cross-reactivity cannot be excluded. Expanded testing with authenticated reference RNAs (DENV1–4, TBEV subtypes, JEV) and clinical specimens is planned.

In addition, an interference study was conducted using three representative materials commonly employed during specimen transport to assess their potential impact on RT-PCR results. To evaluate the influence of specimen transport materials on RT- PCR performance, contrived specimens were prepared by spiking synthetic RNA with four different media: phosphate-buffered saline (PBS), viral transport medium (VTM), EDTA-treated matrix, and a no-additive (neat) condition. Each contrived specimen was formulated at nominal concentrations of 74.5 copies/µL for the *NS2A* target and 70.41 copies/µL for the C target. Normal human serum (Sigma) served as the negative matrix control, synthetic RNA was used as the positive control, and DNase/RNase-free water (Invitrogen) was included as an additional negative control.

Cross-reactivity testing confirmed that the assay exhibited no cross-reactivity with 50 distinct viruses and bacteria (Appendix A). Furthermore, the interference assessment demonstrated that specimens prepared under neat, PBS, EDTA, and VTM conditions showed no statistically significant differences in mean Ct values. Consistent with the *t*-test results (*p* > 0.05), these findings indicate that the presence of these substances did not significantly influence assay performance (Appendix A).

## 4. Discussion

In this study, we developed and validated a newly designed real-time RT-PCR assay targeting the *NS2A* and *C* genes of OHFV. By designing primers and probes based on all available genomic data, inclusivity across reported OHFV lineages was ensured. Compared with previously reported single-target assays [13], the dual-target approach enhances the robustness of the assay, providing an additional safeguard against genomic variation and potential false-negative results. The inclusion of the hRNase P as an IC represents another improvement over earlier protocols, as it enables the verification of nucleic acid extraction and amplification efficiency in every specimen tested [14]. This feature is particularly critical for frontline diagnostic laboratories, where variability in specimen type and quality is unavoidable. Assay performance was confirmed through rigorous validation. Probit analysis defined the LOD for both the *NS2A* and *C* genes [6], while the establishment of indeterminate zones provides a transparent framework for decision-making regarding repeat testing. Furthermore, specificity was verified not only through in silico analysis but also through experimental evaluation against a wide range of viruses and bacteria, including West Nile virus, dengue virus, Zika virus, tick-borne encephalitis virus, and Ebola virus, and particularly by a comprehensive hemorrhagic fever virus panel. No cross-reactivity was observed, underscoring the suitability of the assay for critical public health scenarios. At the time of this study, RNA reference materials for closely related flaviviruses such as Dengue virus and Tick-borne encephalitis virus (TBEV) were not available; therefore, experimental cross-reactivity testing for these viruses could not be performed. This limitation has been clearly acknowledged in this section. Future studies will include additional cross-reactivity evaluations using Dengue, TBEV, and Japanese encephalitis virus (JEV) RNAs to further validate the assay’s specificity against closely related flaviviruses.

Although OHF has never been reported outside endemic regions, the risk of accidental importation remains credible due to the increased volume of international travel and ecological changes affecting tick and rodent populations [1,15]. Integrating OHFV diagnostics into a broader preparedness framework is a concrete measure that non-endemic countries can adopt to mitigate the threat of emerging flaviviruses. This strategy may serve as a model for other nations seeking to strengthen diagnostic readiness against rare but high-consequence pathogens [16,17].

Nonetheless, certain limitations remain. Validation was restricted to synthetic RNA controls due to the absence of clinical OHFV specimens. Therefore, validation using clinical specimens from endemic regions will be essential to establishing assay performance under real-world conditions [18]. Broader evaluation using field-collected samples containing diverse pathogens would further enhance diagnostic confidence [19]. Finally, inter-laboratory ring trials and external quality assurance schemes are needed to establish harmonization [20].

## 5. Conclusions

We present the first standardized OHFV real-time RT-PCR assay system incorporating dual genomic targets, an IC, indeterminate zone definition, and comprehensive cross-reactivity testing. An assay was developed and documented within an institutionally accredited standard protocol, demonstrating how a non-endemic country can proactively prepare for high-consequence pathogen threats. This approach not only strengthens preparedness but also offers a replicable model for collaboration in epidemic preparedness and response.

## Data Availability

The original contributions presented in this study are included in the article/Appendix A. Further inquiries can be directed to the corresponding author.

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
