# Peer review of "Standardized Diagnostic Assays for Omsk Hemorrhagic Fever Virus"

_pathogens, 2025, doi:10.3390/pathogens14111093_

Round 1

Reviewer 1 Report

Comments and Suggestions for Authors

Lee et al., developed and tested two sensitive (70-74 viral copies) single RT-PCR assays for the detection of Omsk hemorrhagic virus. The tests proved to be sensitive and specific enough to use as further diagnostic tests. My main corcern about the paper that the readers can not repeat this assay as very important data (PCR assay annelaing temperature, cycles, elongation time etc. is missig). Such a diagnostic paper must be written in a way which make it possible for the scientific audience to use this assay in their work. The reader with this paper (in this form) can not do that. Instead of lot of not important data in figure and tables teh authors should decribe their method properly.

Abstract

No abbreviations in the Abstract chapter.

Introduction.

Even if it is hidden cited, the authors should indicate, that an RT-PCR system detecting Omsk HFV has already been published (ref 4).

Mat meth.

2.1.1. + 3.1. – Samples. Human, animal? How many? Where from? How many? blood, how many CSF?? No details are added.

line 100 – Why did not the authors developed a nested system, to increase the sensitivity of the assay? What was the size of the PCR products? Even if the authors used real-time PCR system, they should make a classical PCR assay to show the picture of the products with a marker after electrophoresis in a gel.

Are these amplified sequences are conserved or not conserved? Is determination of strains, genotypes possible on the base of the sequences of the PCR produsts?

Results:

lines 173-178 + Table 5 – Why were these threshold values selected?

219-222- The authors should state that these detection limits came from (not correctly described) target, on biological samples the detection limits could change.

line 262 – Crossreactivity studies without testing related Flaviviruses? The authors could get Dengue or TBEV RNA of other samples to test.

Spelling:

line 21 – The detection limits were …….

line 24 – 1 and is enough

line 45 – What is ROK? Has not been written in full before.

line 119 – procedures were …

lines 132-139 – everything in past, not in present

line 180 – consi-dered, or con-sidered

Reviewer 2 Report

Comments and Suggestions for Authors

Materials and Methods

  1. “1.2.11 Specimens” should be corrected to “2.1.1 Specimens.”
  2. Replace all instances of “Ns2a” with “NS2A” (the conventional naming for flavivirus nonstructural proteins). This applies to titles, tables, and figure legends.
  3. In the primers/probes section, the hRNase P probe is labeled as TexaRed…QSY2, whereas in Table 3 it appears as Cy5. The fluorophore and quencher must be unified throughout.
  4. At the end of section 2.2, delete the extra period: “maintained..” → “maintained.”
  5. Add details of the qPCR instrument model, software version, detection channels, and threshold-setting criteria.
  6. Provide the slope, R², amplification efficiency (%), and linear range for both targets; include representative amplification and standard-curve plots.

Results

  1. In the tables, replace the Korean word “Ct 값” with the English “Ct value.”
  2. Verify and clearly describe the re-test/confirmation criteria for Cases 7–10, where only one target was positive.

Round 2

Reviewer 1 Report

Comments and Suggestions for Authors

I accept the reply, answers of the authors, except I still suggest to include the text the answer they gave me about the type/source of target, the sentences they wrote to my question (lines 91-94). 

Author Response

Reviewer Comment: 

I accept the reply, answers of the authors, except I still suggest to include the text the answer they gave me about the type/source of target, the sentences they wrote to my question (lines 91-94). 

Author Response:

We sincerely thank the reviewer for this valuable follow-up suggestion. In line with the reviewer’s recommendation, we have incorporated the additional explanation into the main text. Specifically, the following sentences have been added to lines 85–88 of the revised manuscript:

Although this assay was originally designed for testing human clinical specimens, no human or animal OHFV infections have been reported in Korea to date. Therefore, in the absence of clinical specimens, analytical validation was performed using in vitrotranscribed RNA standards.